# Effects of Ammonification–Steam Explosion Pretreatment on the Production of True Protein from Rice Straw during Solid-State Fermentation

**Bin Li [1],*, Chao Zhao [1], Qian Sun [2], Kunjie Chen [3], Xiangjun Zhao [1], Lijun Xu [1] , Zidong Yang [1] and Hehuan Peng [1]**

1   College of Optical, Mechanical and Electrical Engineering, Zhejiang Agriculture and Forestry University, Hangzhou 311300, China; zhaochao@zafu.edu.cn (C.Z.); zhaoxiangjunzpp@126.com (X.Z.); zoelijun@zafu.edu.cn (L.X.); yzd66@126.com (Z.Y.); peng6907hh@163.com (H.P.)
2   Jiangsu Academy of Agricultural Sciences, Nanjing 210014, China; wendysunjaas@163.com
3   College of Engineering, Nanjing Agricultural University, Nanjing 210031, China; kunjiechen@njau.edu.cn
*   Correspondence: binli@zafu.edu.cn

**Abstract:** It is difficult to obtain high-protein contents from rice straw using direct fermentation due to its low nitrogen content. This study investigates the effects of ammonification–steam explosion pretreatment of rice straw on the protein content after solid-state fermentation (SSF). The pretreatment is carried out under multi-strain inoculation conditions. The samples of rice straw after ammonification ($T_A$), steam explosion ($T_{SE}$), and ammonification and steam explosion ($T_{A-SE}$) were compared to the control group ($T_C$). The results indicate that both ammonification and steam explosion could disintegrate rice straw's lignocellulosic structure, releasing nutrients that can be used for microbial reproduction. In addition, amino compounds are formed along with depolymerization products, thus effectively promoting the true protein content. Post-fermentation, total crude protein contents of $T_A$, $T_{SE}$, and $T_{A-SE}$ samples were 2.56, 1.83, and 4.37 times higher than that of Tc samples, respectively, and true protein contents were 2.52, 1.83, and 5.03 times higher. This study shows that the true protein content by combined ammonification and steam explosion pretreatment of rice straw during 96 h of solid-state fermentation was 46.7% of its total matter, rendering it a suitable alternative to high-protein animal feed.

**Keywords:** steam explosion; lignocellulose; solid substrate fermentation; true protein; rice straw





## 1. Introduction

Plant and animal proteins and single cell protein (SCP) are the main feed protein sources. SCP, also known as microbial or strain protein, is produced by bacteria, fungi, some algae, and other microorganisms when they are supplied with an appropriate substrate and growth conditions [1]. SCP strain components mainly include proteins, carbohydrates, fats, nucleic acids, vitamins, and minerals, as well as a rich source of enzymes and biologically active substances [2]. Therefore, SCP can be used as high-quality protein feed for animals or as food for humans. SCP has a higher yield and a faster production cycle than animal proteins; therefore, it has high growth and transformation rates and is highly efficient [3]. Additionally, SCP can incorporate some agricultural waste products and other raw non-protein materials into protein products, which aligns with the modern green production concept. Therefore, SCP is important for future animal feed and human food development.

Solid-state fermentation (SSF) is a biochemical process used for culturing microorganisms in solid materials. Its main feature is the propagation of microorganisms on solid materials without free water. The water required for fermentation is absorbed from the solid substrate; consequently, aerial hyphae cells mainly perform carbon transport and oxygen uptake [4]. Previous studies have shown that SSF can yield high biomass and

protein content since filamentous fungi secrete a large number of stable enzymes such as cellulases [5]. Biomass and enzyme production are important aspects of SSF that can be optimized to increase the yield and quality of the product. The physical composition of the medium and environmental controls are the most critical factors affecting SSF [6].

Rice straw is a crucial biomass resource and an essential raw material for SSF. China has the world's largest traditional agriculture output, with an annual output of approximately 200 million tons [7]. Currently, more than 70% of the straw in China is left in the field or burned, affecting air quality and human health. Therefore, using rice straw as animal feed can improve climate and health indices. Mature rice straw mainly consists of plant cell wall structural materials. Lignocellulose is the plant cell wall's main component, including cellulose, hemicellulose, and lignin [8]. Although all lignocellulose components are carbohydrates, this material is only useful as a ruminant feed if it does not contain lignin. It is because lignin is a resistant polymer that encapsulates cellulose and hemicellulose, preventing them from being digested by ruminants [9]. Previous studies have attempted to improve straw digestibility by degrading the lignin through physical, chemical, and biological treatments. These treatments include grinding, steam explosion, alkalization, and ammonization [10,11].

Nitrogen content is the most important indicator of feed protein content. The low nitrogen content of rice straw indicates that its protein content is also low. Furthermore, what little protein rice straw contains (no more than 2% true protein) is associated with the lignocellulosic components, which cannot be digested effectively, thereby limiting the usefulness of rice straw as a protein-rich feed [12]. Therefore, regulating the nitrogen flux positively affects protein synthesis during SSF. Inorganic nitrogen is commonly added to feed to improve its crude protein content. However, as inorganic nitrogen is not a true protein, it must still be absorbed and transformed by microorganisms during digestion in livestock and poultry to synthesize true protein [13]. Consequently, the quality of this type of feed is low. Therefore, improving the true protein content of the feed is an important area of research.

Studies have shown that some microorganisms can use inorganic nitrogen to produce SCP. Somda et al. found that SCP production could be improved by using the fungus *Candida* in SSF if a nitrogen source was added [14,15]. It has also been shown that the proper pretreatment of materials before SSF biotransformation improves the protein yield in feed [3,16]. For example, Jalasutram et al. found that *Candida* produced a higher protein yield with an undigested poultry house waste substrate than a digested poultry house waste substrate [17].

Rice straw contains a high concentration of polysaccharides due to the presence of lignocellulosic material, making it an ideal resource for producing SCP. However, if the material is directly fermented in a solid state, it is not easy to achieve efficient biotransformation and a high SCP content within a short time because the lignocellulosic fiber structure causes slow depolymerization [18]. However, previous studies have shown that steam explosion pretreatment can break the chemical association between lignocellulose, thus enabling its effective degradation [19]. The combination of steam explosion and ammonification can achieve a greater degree of lignocellulosic biomass degradation and simultaneous nitrogen enrichment. Therefore, this is a promising method for improving the true protein content of feed during SSF [15].

*Candida prion* can easily transform inorganic nitrogen into organic nitrogen. It can produce many bioactive substances during the fermentation process, making it suitable for the production of SCP feed [19]. *Aspergillus* can produce various enzymes, including hemicellulase, pectinase, protease, amylase, lipase, and tannase [6]. Therefore, combining these two strains can theoretically connect the microbial propagation chain in the SSF process.

Economic decline and the disruption of food production chains due to the COVID-19 pandemic and the war between Russia and Ukraine have exacerbated the fragility of the animal protein supply chain. Using low-cost rice straw as a substrate to produce SCP can

provide an economical and feasible protein source that meets the demands of animal feed protein and is likely to play an important role in promoting the development of the feed industry. However, it remains unclear whether combined physical, chemical, and microbial methods can effectively improve the true protein content of rice straw feed. The impact of ammonification and steam explosion co-treatment on lignocellulose degradation during SSF is unclear, and the quality of the true protein output remains uncertain. Currently, there are no experimental studies regarding these processes. Therefore, this study scientifically and systematically analyzed these processes with the aim of determining their relationships and effects.

## 2. Materials and Methods

### 2.1. Samples

The rice straw used in the experiments was harvested from paddy fields in Meng Cheng County, Anhui Province, China. The straw was pulverized and passed through a 40-mesh sieve before being dried at 105 °C for 2 h. The resulting samples were then used to determine the chemical properties of the rice straw.

### 2.2. Pretreatment Methods

Four sample groups were prepared for the different pretreatments before subjecting them to SSF. The control group ($T_C$) was fermented without pretreatment. The ammonification group ($T_A$) underwent ammonification pre-treatment, the steam explosion group ($T_{SE}$) underwent steam explosion pre-treatment, and the ammonification and steam explosion group ($T_{A-SE}$) underwent both ammonification and steam explosion pre-treatments.

#### 2.2.1. Ammonification

Ammonification was achieved by adding ammonia water to the straw samples at a mass ratio of 1:10 in a 50 L plastic bucket. Distilled water was added to the bucket until the ammonia content was 40%. After even stirring and impregnation, straw samples were sealed and stored at a temperature of 40 °C for 15 days.

#### 2.2.2. Steam Explosion

The $T_{SE}$ device used in this study was the same device used by [9], with a steam explosion pressure of 1.4 MPa maintained for 3 min. The $T_{SE}$ pretreatment of the $T_{A-SE}$ sample was performed 7 d after ammonification.

### 2.3. Solid-State Fermentation

#### 2.3.1. Microorganisms and Cultures

Activated forms of *Aspergillus niger* CICIMF 0410 and *Candida prion* CICC 31949 (China Industrial Microbial Culture Preservation and Management Center) were added to potato dextrose agar (PDA) medium (200.0 g potato, 20.0 g glucose, 20.0 g agar, and 1000 mL distilled water; pH 6.0–6.5). After culturing for 3 days at 28 °C, liquid slanting seed media were created for developing microbial cultures. The *Aspergillus niger* medium contained 0.5 g of rice straw powder, 0.50 g of $MgSO_4 \cdot 7H_2O$, 0.22 g of $K_2HPO_4$, and 100 mL of distilled water (natural pH). The *Candida prion* medium contained 0.5 g of maize powder, 0.50 g of $MgSO_4 \cdot 7H_2O$, 0.2 g of $K_2HPO_4$, and 0.22 g of $KH_2PO_4$ (pH 6.4). These media were then added to a liquid medium with 3 g of maize powder, 0.5 g of rice straw powder, 0.50 g of $MgSO_4 \cdot 7H_2O$, and 0.2 g of $K_2HPO_4$ (neutral pH) and shaken at 120 rpm. The coating plate method was used to dilute the number of microbial colonies, and sterile water was used to adjust the concentration of the fungal strains to $1 \times 10^8$ CFU $mL^{-1}$ [7].

#### 2.3.2. Fermentation

SSF was performed in a 25 L fermentation tank with two tubes in the lid: one tube had a check valve that allowed waste gas to escape, while the other passed sterile air into the flask at a flow rate of 20 $m^3$/h. After drying, 2 kg of each sample was placed into a

fermentation flask with an SSF medium comprising 2.0% $(NH_4)_2SO_4$, 0.05% $MgSO_4$, 0.01% $KH_2PO_4$, and 97.94% rice straw powder (natural pH). All four samples were humidified to 60% and inoculated simultaneously with a mixed fungal culture of Aspergillus niger CICIMF 0410. *Candida tropicalis* CICC 31949. The samples were then inoculated with a 10% (*v*/*w*) mycelium suspension and placed in a sterile constant temperature environment at 35 °C, where they were allowed to ferment for 96 h.

Samples of the fermented liquid were collected after 12, 24, 36, 48, 60, 72, 84, and 96 h of fermentation. These fermented liquid samples were diluted and centrifugally filtered before being analyzed for cellulose and lignin enzymatic activities. The lignocellulose content was determined from the biomass after it was dried. Protein morphology and amino acid fractions were analyzed and compared with the initial values obtained from the control group.

## 3. Analytical Methods

### 3.1. Determination of Lignocellulose Content and Degradation Rate

Neutral detergent fiber (NDF), acid detergent fiber (ADF), and acid detergent lignin (ADL) contents were determined using the Van Soest method [20]. Briefly, neutral soluble ingredients were washed out using a neutral detergent (NDF was insoluble). Acidic soluble ingredients were washed with an acidic detergent (ADF was insoluble). Cellulose was washed out with 72% mass fraction sulfuric acid and the remaining residues of acid insoluble substance (AIS). The AIS was ashified at 550 °C to remove lignin, and the residue was the ash. The differences between the NDF and ADF, ADF and AIS, and ash contents were estimated as the hemicellulose, cellulose, and lignin contents, respectively.

The utilization of lignocellulose during SSF is expressed by the degradation rate, where a higher degradation rate indicates higher utilization. To account for the relative contents of their components, the degradation rates of hemicellulose, cellulose, and lignin were determined using the following formula:

$$R_D = (M_0 - M_x)/M_0 \times 100 \tag{1}$$

where $R_D$ is the degradation rate of each lignocellulose component, and $M_0$ and $M_x$ are each component initial and final contents, respectively. $M_x$ was determined using the following formula:

$$M_x = D_x/D_0 \times c\% \tag{2}$$

where $D_x$ is the measured extracted dry matter content, $D_0$ is the mass of the dry matter in the original sample, and $c\%$ is the percentage content of the cellulose component.

### 3.2. Determination of Enzymatic Activity

Enzyme activity was calculated by adding 10 g of each fermentation sample to 200 mL of distilled water, shaking the mixture at 120 rpm for 30 min, and centrifuging it at $4000\times g$ for 10 min. The resulting supernatant was filtered through a 0.45 µm filter paper. The resulting 1:20 enzyme solution was used to determine the enzymatic activities of cellulose and lignin. Filter paper activity (FPA) was measured using the 3,5-dinitro salicylic acid (DNS) method. The reducing sugars produced by the hydrolysis of filter paper produce red–brown amino compounds with DNS, with maximum light absorption at 540 nm. The color depth of the reaction solution is proportional to the reducing sugar content, which allows the enzymatic activity of the filter paper to be calculated. A 1.0 mL sample of the diluted solution was added to 1.0 mL of 0.1 mol/L acetic acid buffer (pH 4.6) and pre-heated to 50 °C, after which 50 mg of quantitative filter paper (1 cm × 6 cm) was added and maintained at 50 °C for 1 h. Then, 2 mL of a 10% NaOH solution was added to stop the enzymatic reaction immediately. After the sample cooled, the absorbance was measured at a wavelength of 540 nm using a UV-visible spectrophotometer (759S Shanghai Jing Hua Technology Instrument Co., Ltd., Shanghai, China). Only the diluted enzyme liquid and

acetic acid buffer were added to the control group. A blank group containing the filter paper and the buffer was also analyzed.

CMCase activity was measured as follows. A 0.5 mL sample of the diluted enzyme solution was added to a reaction substrate of 1.5 mL of 0.5% CMC-Na solution and 1.5 mL of the acetic acid buffer solution. The sample was maintained at 50 °C for 30 min and then terminated with 2 mL of a 10% NaOH solution. The reduced sugar content was then determined using the colorimetric method at 540 nm.

Enzymatic activity was defined as one unit of cellulase enzyme that released 1 μmol of glucose equivalent per minute at a pH of 4.8 in a 50 °C water bath: [21]

$$Enzyme\ activity\ (IU/mL) = \frac{C \times V_s \times n}{V_t \times t} \tag{3}$$

where $C$ is the amount of released reducing sugar ($\mu mol/mL$), $V_s$ is the reaction system volume ($mL$), $n$ is the dilution factor, $V_t$ is the test volume ($mL$), and $t$ is the reaction time (min).

The lignin peroxidase (LiP) activity was determined using the oxidation of veratryl alcohol to veratraldehyde in the presence of hydrogen peroxide. A sample with a total reaction volume of 3 mL was used to determine the LiP activity. The sample contained 1.85 mL of 0.125 mol/L sodium tartrate buffer (pH 3.0), 0.1 mL of 10.0 mmol/L veratryl alcohol, and 1.0 mL of crude enzyme solution. The sample was pre-heated to 37 °C, and 0.05 mL of 20 mmol/L $H_2O_2$ was added to start the reaction. Absorbance was measured at 310 nm over 2 min. The change in optical density (OD) per minute was 0.1% of 1 enzyme activity unit (U):

$$Enzyme\ activity\ (IU/mL) = \frac{\Delta OD \times V_s \times d}{\varepsilon \times V_t \times t} \tag{4}$$

$\Delta OD$ is the change in optical density, $\varepsilon$ is the molar extinction coefficient of veratraldehyde (L/mol/cm), $Vs$ is the reaction system volume ($mL$), $Vt$ is the sample volume ($mL$), d is the cuvette light diameter (1 cm), and $t$ is the reaction time (min).

For the LiP activity analyses, $\varepsilon$ was set to 9300 L/mol/cm, $V_s$ was 3 $mL$, and $V_t$ was 0.5 mL.

Laccase activity was measured as follows. The oxidation of 2,2′-Azinobis-(3-ethylbenzthiazoline-6-sulphonate) (ABTS) was used to represent the activity of laccase (Lac) in a sample with a total reaction volume of 4 mL. This sample contained 2 mL of 0.5 mmol/L ABTS (dissolved in 0.1 mol/L acetic acid-sodium acetate buffer at pH 5.0) and 2 mL of crude enzyme solution. The mixture was placed in a 4 mL colorimetric dish, and the change in absorbance at 420 nm was determined over 3 min. One enzyme activity unit (μmol/(L·min)) was defined as the amount of enzyme required to transform 1 μmol of ABTS per minute and was calculated using Equation (4) ($\varepsilon$ = 3600 L/mol/cm, $V_s$ = 4 mL, and $V_t$ = 0.5 mL).

### 3.3. Determination of Chemical Composition

The reducing sugar contents were determined using the DNS method, and total nitrogen (TN) was determined using the Kjeldahl method [22]. Available nitrogen was determined using the alkali decomposition diffusion method. The samples were hydrolyzed in a sealed dispersion dish with a 1.8 mol/L sodium hydroxide (NaOH) solution. The available nitrogen was converted to ammonia, continuously dispersed and released, absorbed by boric acid ($H_3BO_3$), and then titrated with hydrochloric acid. The amount of available nitrogen was calculated from the hydrochloric acid used [8].

The protein content was determined using the semi-trace Kjeldahl method [20]. A total of 30 g of each fermentation sample was centrifuged at 10,000× *g* for 5 min. The solids obtained after centrifuging were washed with distilled water thrice and dried at 65 °C until they reached a constant weight. The washing solution was collected, and the true protein was analyzed using colorimetry under 280 and 260 nm standard protein (to avoid the added soluble nitrogen source). The true protein content of the solid material

was determined using the tungstic acid deposition method, in which 8 mL of a 10% sodium tungstate solution was cultured at 25 °C for 30 min. The pH was adjusted to 2.0 by adding 10 mL of 0.5 mol/L sulfuric acids. The culture tube was left overnight at room temperature (10–20 °C). The filtrate residue was rinsed with distilled water, and its nitrogen content was determined using the Kjeldahl method. The true protein content was 6.25 times the nitrogen content.

The composition of amino acids in the proteins was determined using an 835-50 model automatic amino acid analyzer (Hitachi Co, Ltd., Hitachi, Japan). The analyzer was hydrolyzed with 30 mL of 6 mol/L HCl and filtered through filter paper, after which 50 μL of the sample was injected. A standard analysis was then performed using a sodium citrate system. Each sample was analyzed in triplicate.

### 3.4. Statistical Analyses

Analytical results are given as mean values ± one standard deviation. Statistical analyses were performed using the SPSS software package (SPSS 18.0, International Business Machines Corporation, New York, NY, USA), and the significance of the experimental results was evaluated using one-way analysis of variance (ANOVA) and Duncan's multiple range tests ($p \leq 0.05$).

## 4. Results and Discussion

### 4.1. Effects of Pretreatment on Lignocellulosic Compositions

As shown in Table 1, fermentation of the $T_C$ sample (no pretreatment) resulted in a lignocellulose content of 70.4% and a soluble sugar content of only 0.90% from direct hydrolysis. According to these results, rice straw should be categorized as roughage. The crude protein content of the $T_C$ sample accounted for 4.48% of the dry matter mass, indicating that rice straw is not a high-quality protein feed. However, after ammonia pretreatment ($T_A$), the hemicellulose and cellulose contents of the rice straw decreased to 20.2% and 33.3%, respectively, while the lignin content increased slightly to 9.96%. Meanwhile, the soluble reducing sugar and nitrogen contents also increased substantially compared to the $T_C$ sample (9.4 and 2.5 times, respectively). The $T_{SE}$ sample also had lower hemicellulose and cellulose contents than $T_C$ (17.1% and 35.9%, respectively), and a higher lignin content (9.50%). The soluble nitrogen to TN ratio in the $T_{SE}$ sample (0.52) also increased compared to $T_C$ (0.03). The $T_{A-SE}$ sample had the lowest hemicellulose content of all four samples (15.8%); however, there was no significant difference in the cellulose content (33.3%) compared to $T_A$ and $T_{SE}$. Furthermore, $T_{A-SE}$ had the lowest lignin content (9.15%) of the three pre-treated samples, which was slightly higher than $T_C$. The $T_{A-SE}$ sample also had the highest TN and soluble nitrogen contents (2.45% and 1.43%, respectively) of all four samples. The data detection bias did not influence the results of different samples. Differences in the $T_A$, $T_{SE}$, and $T_{A-SE}$ compositions compared to $T_C$ were due to pretreatment. The analyzer was hydrolyzed with 30 mL of 6 mol/L HCl and filtered through filter paper. Then, 50 μL of the sample was injected, and a standard analysis was performed using a sodium citrate system. Each sample was analyzed in triplicate [15]. This process increases the hemicellulose content in the dry matter. As hemicellulose is easily hydrolyzed, its content in solid dry matter decreases, but its content in the liquid increases. Nitrogen fixation also occurs on some of the lignin hydroxyl and carboxyl groups during ammonification, increasing nitrogen content [23].

Steam explosion degrades cellulose and lignin by destroying the hydrogen bonds between the cellulose and hemicellulose and causes acetyl hydrolysis at high temperatures. This results in lignin's softening, making it sticky when the water vapor temperature is higher than the glass transition temperature of lignin; it helps to degrade the lignin, even though the high tear strength directly interrupts cellulose on the main chains of C–O and C–C bonds [24]. As a result, the binding energies between lignin and cellulose and between lignin and hemicellulose are overcome [9,20]. The low lignin content in $T_{A-SE}$ compared to those of $T_A$ and $T_{SE}$ may be due to pseudo-lignin production during the steam explosion.

Fracturing the ether bonds in lignin enhances their reactivity, forming humic acids with degraded hemicellulose [25].

**Table 1.** Effects of pretreatment on sample composition (%).

| Sample | Hemicellulose | Cellulose | Lignin | Reducing Sugar | Total Nitrogen | Soluble Nitrogen | Crude Protein | True Protein | *p*-Value |
|---|---|---|---|---|---|---|---|---|---|
| $T_C$ | 23.6 ± 0.08 [a] | 37.7 ± 0.11 [a] | 9.12 ± 0.15 [c] | 0.90 ± 0.05 [d] | 0.72 ± 0.08 [c] | 0.02 ± 0.01 [c] | 4.48 ± 0.12 [c] | 1.67 ± 0.03 [c] | 0.05 |
| $T_A$ | 20.2 ± 0.05 [a] | 33.3 ± 0.12 [b] | 9.96 ± 0.16 [a] | 8.46 ± 0.04 [c] | 1.83 ± 0.10 [b] | 0.75 ± 0.02 [b] | 11.4 ± 0.15 [b] | 1.92 ± 0.04 [b] | |
| $T_{SE}$ | 17.1 ± 0.07 [b] | 35.9 ± 0.13 [b] | 9.50 ± 0.16 [a] | 9.52 ± 0.04 [a] | 0.73 ± 0.11 [c] | 0.38 ± 0.02 [c] | 4.51 ± 0.09 [c] | 2.26 ± 0.05 [a] | |
| $T_{A-SE}$ | 15.8 ± 0.06 [b] | 33.3 ± 0.12 [b] | 9.15 ± 0.17 [b] | 5.49 ± 0.05 [b] | 2.45 ± 0.09 [a] | 1.43 ± 0.06 [a] | 15.3 ± 0.16 [a] | 2.43 ± 0.05 [a] | |

Note: $T_C$, control group; $T_A$, ammonification; $T_{SE}$, steam explosion; $T_{A-SE}$, ammonification and steam explosion. All data were presented as means plus/minus standard error. The letters following the numbers in the same column represent significant differences at the level of 0.05 by Duncan's multiple comparison testing.

The results indicate that pretreatment had a substantial influence on the lignocellulosic composition.

### 4.2. Dynamic Changes in Enzyme Activity

Cellulase Activity

Unlike the chemical and physical degradation methods mentioned above, SSF achieves biotransformation using enzymes secreted by microorganisms. These enzymes act on a solid substrate to break the chemical bonds in polymers, including cellulose and lignin, thereby depolymerizing the entire structure [6]. Enzyme activity reflects the microorganisms' growth performance and the enzymes' depolymerization ability.

In Figure 1, it can be seen that the carboxymethyl cellulase (CMCase) activity and filter paper activity (FPA) varied over time in all four sample groups. The maximum CMCase activities were observed for $T_C$, $T_A$, $T_{SE}$, and $T_{A-SE}$ samples at 2.76, 3.06, 3.56, and 3.85 U· $mL^{-1}$, respectively, after 60, 48, 36, and 36 h of fermentation, respectively. The maximum FPA activities were observed for $T_C$, $T_A$, $T_{SE}$, and $T_{A-SE}$ samples at 0.61, 0.93, 1.34, and 1.69 $U·mL^{-1}$, respectively, after 72, 60, 48, and 36 h of fermentation, respectively.

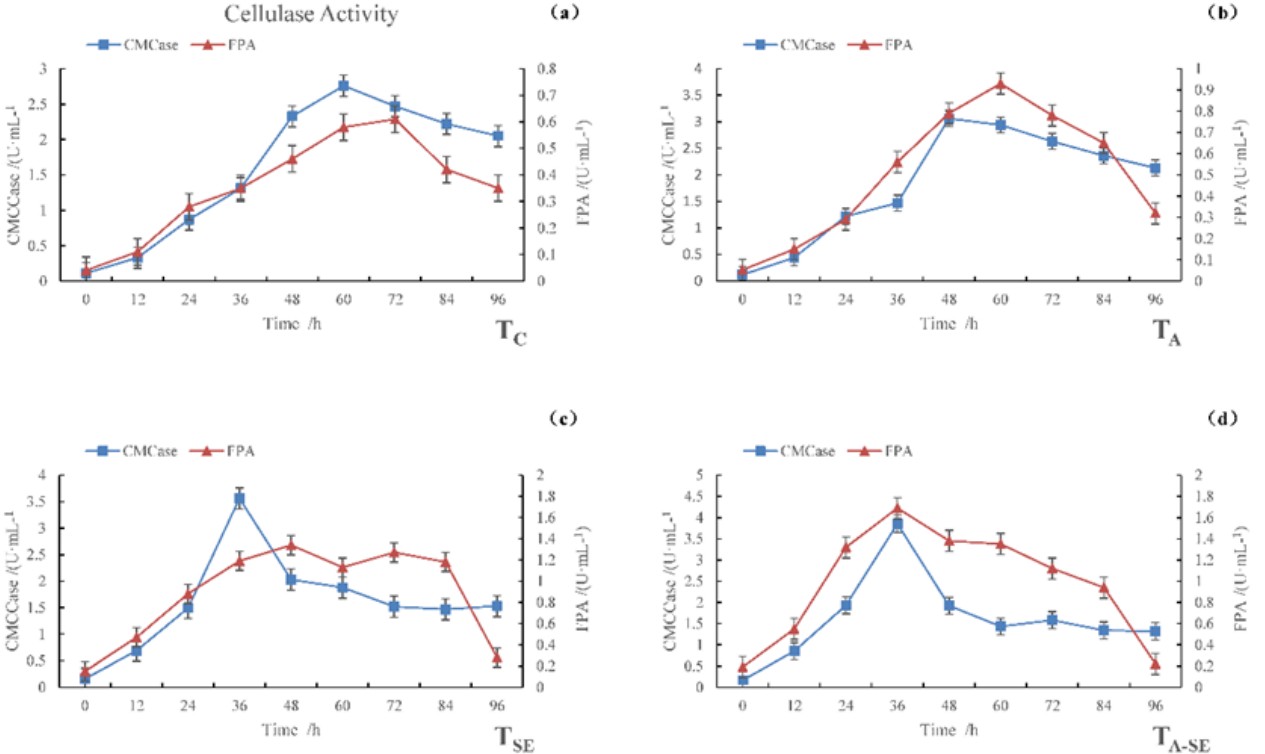

**Figure 1.** CMCase and FPA activities during the SSF process. (**a**) $T_C$, (**b**) $T_A$, (**c**) $T_{SE}$, and (**d**) $T_{A-SE}$.

Cellulose is the main carbohydrate in lignocellulosic cell walls. Hemicelluloses are amorphous, branched, single-stranded polyphasic polysaccharides that attach to adjacent cellulose fibers via non-covalent cross-linking. Lignin is a complex cross-linked polymer that plays an important role in binding lignocellulosic substrates and strengthening cell walls. The amounts and structures of these three main components largely depend on the type of lignocellulosic biomass. The degree to which cellulose and hemicellulose in the biomass are linked to lignin through hydrogen and covalent bonds contributes substantially to the degradation resistance of the biomass, which affects the enzymatic hydrolysis process [8].

In natural cellulose materials, hemicellulose and lignin tightly wrap cellulose, making it difficult for cellulase to contract and adsorb on the cellulosic substrate. This is an issue for fermentation, as cellulase's main factor affecting its hydrolysis efficiency is its cellulose accessibility. The enzymatic hydrolysis efficiency of natural cellulosic material is typically low. Therefore, proper pretreatment before enzymatic hydrolysis can remove the hemicellulose and lignin surrounding the cellulose, thereby improving the specific surface area of the fiber. It allows the enzymes to access cellulose more easily [9], enhancing the microorganisms' reproductive performance and cellulase activity.

Degradants, including glucose and cellobiose, inhibit the synthesis of cellulase. In contrast, the catalytic function of cellulase is inhibited by the feedback of the end products of the enzymatic reaction. The sugars produced by enzymes in the mixed fermentation system are quickly utilized by yeast strains, thus avoiding any inhibitory effects caused by sugar accumulation. It allows the enzymes to decompose the substrate efficiently. In addition, the sugar produced promotes yeast growth.

*4.3. Lignin Enzymes*

Figure 2 shows that, during the fermentation of all four samples, the activities of the two ligninase enzymes initially increased, then decreased. The maximum LiP enzyme activities of the $T_C$, $T_A$, $T_{SE}$, and $T_{A-SE}$ samples were 186, 545, 806, and 1517 $U \cdot mL^{-1}$, respectively, and occurred after 72, 48, 36, and 36 h of fermentation, respectively. The maximum Lac activities of the $T_C$, $T_A$, $T_{SE}$, and $T_{A-SE}$ samples were 51, 183, 233, and 332 $U \cdot mL^{-1}$, respectively, and occurred after 72, 60, 48, and 36 h of fermentation, respectively. These results indicate that the distinct pretreatment methods differentially affected the lignin enzyme activities. The $T_{SE}$ sample underwent more enzymatic hydrolysis than the $T_A$ sample, while both underwent much more hydrolysis than the $T_C$ sample. However, the $T_{A-SE}$ sample had the highest LiP and Lac enzyme activities, which were 8.2 and 6.5 times larger than the control group, respectively. The time to reach peak activity was also substantially shorter in all three pre-treated samples than in $T_C$. In $T_C$, Lac reached its maximum activity before LiP. However, in the $T_A$ and $T_{SE}$ samples, the LiP enzyme reached its peak activity before Lac, while in the $T_{A-SE}$ sample, both enzymes reached their peak activities at approximately the same time.

The increase in ligninase activity after pretreatment likely occurred because fungi play a key role in lignin degradation. During the initial stage of microbial decomposition, fungal spores and mycelium can diffuse into lignin. Microorganisms secrete LiP, Lac, and manganese peroxidase (MnP) to change the properties of lignin. Subsequent enzymatic catalysis starts a chain reaction of free radicals, resulting in lignin degradation [26]. These fast and slow sequences occur because of extracellular microbial enzymes' different enzymatic hydrolysis mechanisms. As a result, the different pretreatment methods resulted in different lignin disintegration characteristics. In addition, glucose exerts a non-competitive inhibitory effect on cellulase, and the use of mixed strains affects the interaction and sequence of lignocellulosic degradation, which, in turn, affects the rate and extent of degradation.

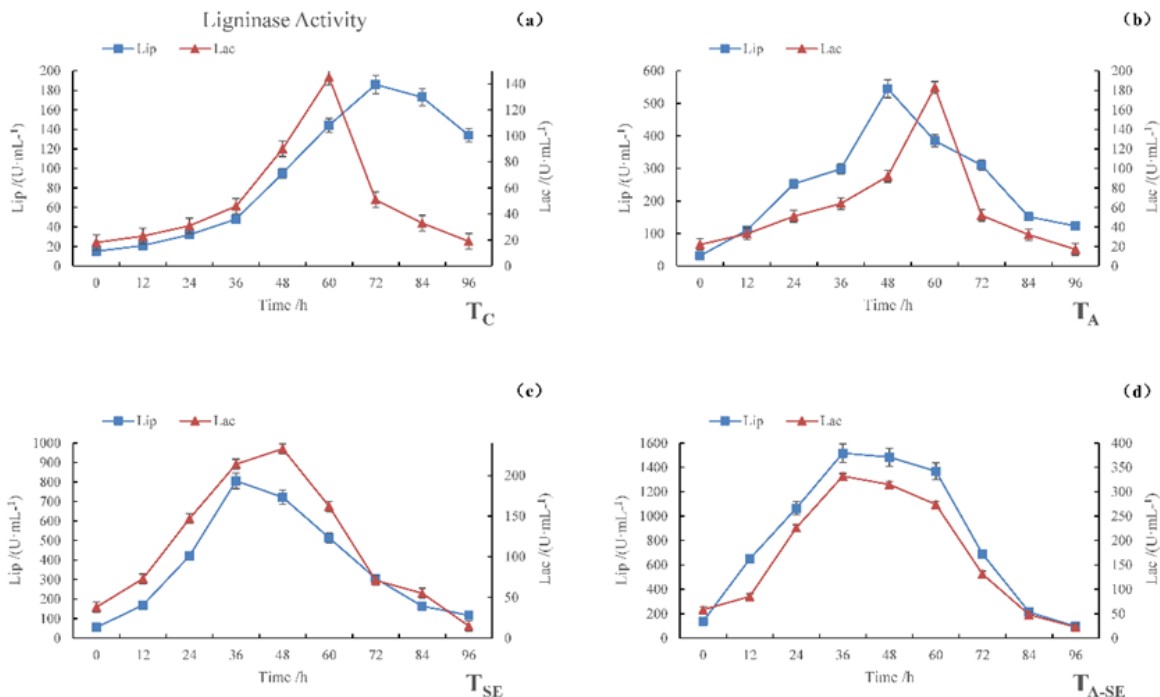

**Figure 2.** LiP and Lac activities during the SSF process. (**a**) $T_C$, (**b**) $T_A$, (**c**) $T_{SE}$, and (**d**) $T_{A\text{-}SE}$.

*4.4. Nutrient Utilization*

The chemical compositions of the samples after 96 h of fermentation are listed in Table 2. The data indicate that, in all four samples, multi-strain SSF significantly degraded the hemicellulose, followed by cellulose and lignin. The rate of dry matter weight loss can explain differences in the compositions of the groups. During SSF, the material lost weight, owing to the growth and metabolism of microorganisms [27]. The more vigorous the microorganism growth and reproduction, the more substantial the weight loss. The TA-SE sample showed the smallest decrease in dry matter sugar content after the reaction, followed by the TSE and TA groups. This indicates that microbial growth and consumption of available sugar in the substrate occurred, as the reducing sugar produced by SSF was gradually consumed by the microbes over time.

**Table 2.** Sample compositions after 96 h of SSF (%).

| Sample | Hemicellulose | Cellulose | Lignin | Reducing Sugar | Total Nitrogen | Soluble Nitrogen | Weight Loss Ratio | *p*-Value |
|---|---|---|---|---|---|---|---|---|
| $T_C$ | 9.31 ± 0.11 [a] | 22.01 ± 0.25 [a] | 8.41 ± 0.23 [a] | 4.32 ± 0.08 [a] | 0.92 ± 0.08 [b] | 0.02 ± 0.01 [d] | 16.7 ± 0.3 [d] | 0.05 |
| $T_A$ | 5.03 ± 0.08 [b] | 15.80 ± 0.21 [b] | 7.20 ± 0.22 [b] | 3.21 ± 0.07 [a] | 1.83 ± 0.10 [a] | 0.75 ± 0.02 [b] | 25.2 ± 0.4 [c] | |
| $T_{SE}$ | 2.52 ± 0.03 [c] | 13.06 ± 0.19 [c] | 5.51 ± 0.16 [b] | 1.58 ± 0.03 [b] | 0.73 ± 0.11 [b] | 0.38 ± 0.02 [c] | 47.8 ± 0.5 [b] | |
| $T_{A\text{-}SE}$ | 0.76 ± 0.01 [c] | 6.85 ± 0.12 [d] | 4.28 ± 0.13 [c] | 0.03 ± 0.01 [c] | 2.45 ± 0.09 [a] | 1.43 ± 0.06 [a] | 64.5 ± 0.5 [a] | |

Note: $T_C$, control group; $T_A$, ammonification; $T_{SE}$, steam explosion; $T_{A\text{-}SE}$, ammonification and steam explosion. All data were presented as means plus/minus standard error. The letters following the numbers in the same column represent significant differences at the level of 0.05 by Duncan's multiple comparison testing.

The hemicellulose, cellulose, and lignin contents of the $T_C$, $T_A$, $T_{SE}$, and $T_{A\text{-}SE}$ samples decreased after SSF. The degradation rates of these three compounds are shown in Figure 3. $T_C$ had the lowest degradation rate; however, physical and chemical pretreatments ($T_A$, $T_{SE}$, and $T_{A\text{-}SE}$) improved the degradation rate by enhancing the effects of the biological

treatment, indicating that pretreatment can comprehensively change the enzymatic activity of cellulose and lignin. The combination of ammonification and steam explosion pretreatments before the fermentation process resulted in almost 100% lignocellulosic decomposition, which can maximize the nutrients available for microbial reproduction.

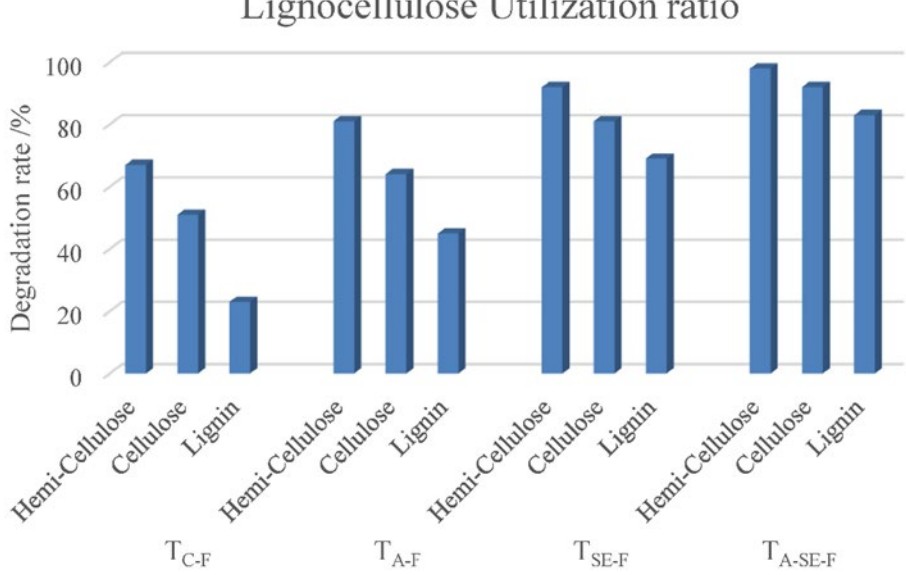

**Figure 3.** Degradation rate as an indicator of the lignocellulose utilization ratio during the SSF process for $T_{C-F}$, $T_{A-F}$, $T_{SE-F}$, and $T_{A-SE-F}$.

### 4.5. Proteins and Amino Acids

4.5.1. Protein Production

The crude and true protein contents of the four SSF samples are listed in Tables 3 and 4, respectively. After fermentation for 96 h, the liquid contained more protein than the solid. The true protein contents of the liquid and solid $T_{A-SE}$ samples were 15.4% and 31.3% of the total weight, respectively, which were 7.2 times larger than the protein content of the liquid $T_C$ samples and 4.4 times larger than the protein content of the solid $T_C$ samples.

The SSF process continuously exchanges energy and matter with the environment. Theoretically, the nitrogen metabolism reaction in this system involves the conversion of different forms of nitrogen-containing compounds in the system and involves three main reaction modes: (1) Assimilation, which re-synthesizes strain proteins using non-protein nitrogen sources (such as $NH_4^+$) during strain growth, is the only way to achieve net protein growth in a reaction system. (2) Transformation, in which a raw protein nitrogen source reacts to synthesize active protein during thallus growth. (3) Decomposition, in which the enzymatic degradation and de-ammonification of protein or protein-like substances occur due to oxidation, energy use, or as a carbon frame. Functional groups carry out organic chemical reactions, while microorganisms can release lignin degradation enzymes that depolymerize lignin macromolecules. Small soluble organic matter components can be directly absorbed, while the lignocellulosic crystal structure produces lignin derivatives [28]. For example, the cellulose chain becomes shorter during the degradation of phenols, quinones, and other products, and oxidation and derivative reactions produce amino derivatives [23]. These derivatives provide a bridge for decomposition reactions and assimilation and play a significant role in SCP fermentation [29].

**Table 3.** Crude protein contents of the $T_C$, $T_A$, $T_{SE}$, and $T_{A-SE}$ samples after SSF.

| Sample | Crude Proteins in Dry Matter (%) | | | | | *p*-Value |
| | Fermentation Time (h) | | | | | |
| | 0 | 24 | 48 | 72 | 96 | |
|---|---|---|---|---|---|---|
| $T_C$ | 4.48 ± 0.15 [c] | 5.36 ± 0.14 [c] | 6.48 ± 0.18 [c] | 7.16 ± 0.18 [c] | 3.47 ± 0.14 [d] | 0.05 |
| $T_A$ | 11.4 ± 0.21 [b] | 10.3 ± 0.25 [b] | 9.86 ± 0.21 [b] | 8.57 ± 0.19 [b] | 8.20 ± 0.18 [b] | |
| $T_{SE}$ | 4.51 ± 0.16 [c] | 5.32 ± 0.16 [c] | 6.49 ± 0.19 [c] | 7.25 ± 0.16 [c] | 7.51 ± 0.17 [c] | |
| $T_{A-SE}$ | 15.3 ± 0.22 [a] | 15.6 ± 0.28 [a] | 16.6 ± 0.27 [a] | 17.6 ± 0.28 [a] | 17.7 ± 0.30 [a] | |

| Sample | Crude Proteins in Liquid (%) | | | | | *p*-Value |
| | Fermentation Time (h) | | | | | |
| | 0 | 24 | 48 | 72 | 96 | |
|---|---|---|---|---|---|---|
| $T_C$ | 0.03 ± 0.01 [c] | 2.57 ± 0.08 [d] | 4.69 ± 0.10 [c] | 6.38 ± 0.13 [d] | 8.91 ± 0.19 [c] | 0.05 |
| $T_A$ | 0.05 ± 0.02 [c] | 6.41 ± 0.12 [c] | 11.1 ± 0.15 [b] | 18.9 ± 0.25 [b] | 23.5 ± 0.34 [b] | |
| $T_{SE}$ | 0.13 ± 0.02 [b] | 8.56 ± 0.11 [b] | 11.1 ± 0.16 [b] | 14.3 ± 0.22 [c] | 15.1 ± 0.22 [b] | |
| $T_{A-SE}$ | 0.22 ± 0.03 [a] | 10.3 ± 0.15 [a] | 25.7 ± 0.26 [a] | 32.1 ± 0.27 [a] | 36.3 ± 0.28 [a] | |

Note: $T_C$, control group; $T_A$, ammonification; $T_{SE}$, steam explosion; $T_{A-SE}$, ammonification and steam explosion. All data were presented as means plus/minus standard error. The letters following the numbers in the same column represent significant differences at the level of 0.05 by Duncan's multiple comparison testing.

**Table 4.** True protein contents of the $T_C$, $T_A$, $T_{SE}$, and $T_{A-SE}$ samples after SSF.

| Sample | True Proteins in Dry Matter (%) | | | | | *p*-Value |
| | Fermentation Time (h) | | | | | |
| | 0 | 24 | 48 | 72 | 96 | |
|---|---|---|---|---|---|---|
| $T_C$ | 1.66 ± 0.09 [a] | 1.82 ± 0.12 [d] | 1.94 ± 0.13 [d] | 2.03 ± 0.12 [d] | 2.15 ± 0.13 [d] | 0.05 |
| $T_A$ | 1.68 ± 0.11 [a] | 3.06 ± 0.17 [b] | 4.92 ± 0.21 [b] | 5.81 ± 0.18 [b] | 6.81 ± 0.23 [b] | |
| $T_{SE}$ | 1.73 ± 0.12 [a] | 2.87 ± 0.16 [c] | 3.24 ± 0.16 [c] | 3.80 ± 0.17 [c] | 4.21 ± 0.24 [c] | |
| $T_{A-SE}$ | 1.82 ± 0.15 [a] | 6.81 ± 0.18 [a] | 11.8 ± 0.24 [a] | 14.5 ± 0.22 [a] | 15.4 ± 0.28 [a] | |

| Sample | True Proteins in Liquid (%) | | | | | *p*-Value |
| | Fermentation Time (h) | | | | | |
| | 0 | 24 | 48 | 72 | 96 | |
|---|---|---|---|---|---|---|
| $T_C$ | 0.01 ± 0.01 [c] | 1.65 ± 0.06 [d] | 3.37 ± 0.18 [c] | 5.68 ± 0.19 [d] | 7.13 ± 0.21 [c] | 0.05 |
| $T_A$ | 0.03 ± 0.01 [b] | 4.56 ± 0.23 [b] | 8.91 ± 0.27 [b] | 13.3 ± 0.26 [b] | 16.6 ± 0.28 [b] | |
| $T_{SE}$ | 0.03 ± 0.01 [b] | 3.67 ± 0.21 [c] | 9.05 ± 0.26 [b] | 11.5 ± 0.31 [c] | 12.8 ± 0.32 [c] | |
| $T_{A-SE}$ | 0.05 ± 0.01 [a] | 12.4 ± 0.30 [a] | 20.1 ± 0.32 [a] | 28. 8 ± 0.35 [a] | 31.3 ± 0.37 [a] | |

Note: $T_C$, control group; $T_A$, ammonification; $T_{SE}$, steam explosion; $T_{A-SE}$, ammonification and steam explosion. All data were presented as means plus/minus standard error. The letters following the numbers in the same column represent significant differences at the level of 0.05 by Duncan's multiple comparison testing. The total mass of the liquid and solid states is 1, and the data in the table are percentage content.

### 4.5.2. Amino Acid Compositions

The amino acid contents of the solid substrate in the $T_{A-SE}$ sample after SSF are listed in Table 5. Combined, all the amino acids comprised 42.1% of the dry matter, 9.8 times that of the $T_C$ sample (4.30%). Eighteen amino acids were observed in the $T_{A-SE}$ solid substrate, including all essential amino acids [30]. The results indicate that, compared with individual treatments, the combination of physical and chemical pretreatments (ammonification and steam explosion) promoted the formation of amino acids during the biological SSF process. Notably, the conditions used for the $T_{SE}$ pretreatment in this study were based on the previously optimized steam explosion conditions used for rice straw feed, ensuring a good taste for animals. These may not be optimal conditions for cultivating microorganisms by SSF for SCP; therefore, future research should focus on the specific steam explosion and ammonification parameters for optimal cultivation. In the next step, GCMS could be used

to analyze the product's specific components, detect whether it can be used as human food, and use low-value biomass resources to produce high-value energy products [31].

**Table 5.** Percentages of amino acids in the $T_{A-SE}$ and $T_C$ solid substrate after SSF.

| Amino Acid Species | Content (%) | | Amino Acid Species | Content (%) | | *p*-Value |
|---|---|---|---|---|---|---|
| | $T_{A-SE}$ | $T_C$ | | $T_{A-SE}$ | $T_C$ | |
| aspartic acid | $4.62 \pm 0.15$ | $0.40 \pm 0.02$ | isoleucine | $1.57 \pm 0.07$ | $0.31 \pm 0.02$ | 0.05 |
| threonine | $1.82 \pm 0.07$ | $0.22 \pm 0.01$ | leucine | $4.31 \pm 0.11$ | $0.36 \pm 0.02$ | |
| serine | $1.46 \pm 0.05$ | $0.23 \pm 0.01$ | tyrosine | $3.55 \pm 0.12$ | $0.15 \pm 0.01$ | |
| glutamic acid | $5.07 \pm 0.18$ | $0.54 \pm 0.03$ | phenylalanine | $2.79 \pm 0.15$ | $0.28 \pm 0.01$ | |
| glycine | $2.15 \pm 0.06$ | $0.26 \pm 0.01$ | lysine | $3.56 \pm 0.11$ | $0.22 \pm 0.01$ | |
| alanine | $0.56 \pm 0.03$ | $0.27 \pm 0.01$ | histidine | $1.20 \pm 0.05$ | $0.13 \pm 0.01$ | |
| cystine | $0.37 \pm 0.02$ | - | tryptophan | $0.03 \pm 0.01$ | - | |
| valine | $2.74 \pm 0.06$ | $0.33 \pm 0.02$ | arginine | $3.09 \pm 0.05$ | $0.19 \pm 0.01$ | |
| methionine | $1.87 \pm 0.06$ | $0.02 \pm 0.01$ | proline | $1.35 \pm 0.04$ | $0.38 \pm 0.02$ | |

Note: $T_C$, control group; $T_A$, ammonification; $T_{SE}$, steam explosion; $T_{A-SE}$, ammonification and steam explosion. All data were presented as means plus/minus standard error - indicates that the ingredient is not detected. The letters following the numbers in the same column represent significant differences at the level of 0.05 by Duncan's multiple comparison testing.

## 5. Conclusions

The combination of amination and steam explosion pretreatment has the ability to rapidly degrade the lignocellulosic components of rice straw, thereby accelerating the subsequent SSF process. Pretreatment increases the output of microbial protein content up to a maximum of 46.7% of the rice straw source matter and shortens the fermentation time. The product contains a range of amino acids, making it suitable for animal use as a high-protein alternative feed. Rice straw is a significant waste product of agriculture in China that is commonly discarded or burnt, which can adversely affect the climate and human health. However, by applying appropriate pretreatment techniques, waste rice straw can be made into cheap viable animal feed. It can thus reduce the detrimental effects on the environment and bolster the currently fragile supply chain of animal proteins.

**Author Contributions:** B.L.: Corresponding author, Conceptualization, Writing, Editing, and Funding acquisition; C.Z.: Investigation, Review, Methodology; Q.S.: Writing original draft; X.Z.: Formal analysis, Review; L.X.: Investigation, Review; Z.Y.: Validation, Review; K.C.: Methodology, Investigation; H.P.: Data curation, Review. All authors have read and agreed to the published version of the manuscript.

**Funding:** This work was supported by the Commonweal Project of Science and Technology Agency of Zhejiang Province of China (LGN22C130201), the Natural Science Foundation of Zhejiang Province (LQ17C130001) and the National Natural Science Foundation of China (31801317).

**Institutional Review Board Statement:** No ethical approval is required for this study.

**Informed Consent Statement:** Not applicable.

**Data Availability Statement:** The data presented in this study are available on request from the corresponding author.

**Acknowledgments:** We would like to thank Farman Ali Chandio (Sindh Agricultural University, Pakistan) for English language editing.

**Conflicts of Interest:** The authors declare no conflict of interest.

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
