# Peer review of "Effects of Ammonification–Steam Explosion Pretreatment on the Production of True Protein from Rice Straw during Solid-State Fermentation"

_sustainability, doi:10.3390/su15075964_

Round 1
Reviewer 1 Report
The authors conducted an experiment to compare the lignocellulosic composition of rice straw with and without pre-treatment, using different methods such as fermentation, ammonia pre-treatment, and steam explosion. The results showed that untreated rice straw had a high lignocellulose content of 70.4% and low soluble sugar content of 0.90%, indicating that it should be categorized as roughage. The crude protein content was also low at 4.48%, suggesting that it is not a high-quality protein feed.
However, after ammonia pre-treatment, the hemicellulose and cellulose contents decreased while the lignin content increased slightly. The soluble reducing sugar and nitrogen contents also increased substantially compared to the untreated sample. Similarly, steam explosion caused the degradation of cellulose and lignin by destroying hydrogen bonds, resulting in the low lignin content in the TA-SE sample
Please describe "what next". Further studies can be performed using GCMS. Please compare with and cite the article: https://doi.org/10.3390/pr9020364
Overall, the text is informative and well-organized, with clear explanations of the experimental results and mechanisms behind them. The use of technical terms and abbreviations may be challenging for readers who are not familiar with the field. However, the authors provide adequate context to help readers understand the content.
Please correct the editing in the article. In this form, I advise against publishing.
Reviewer 2 Report
Comments and Suggestions for Authors
The authors of this paper investigate the effects of ammonification– steam explosion pretreatment of rice straw under multi-strain inoculation conditions on the protein content after solid-state fermentation (SSF). The objective of this study was to determine the relationships and effects of solid-state fermentation processes.
The authors demonstrated that the combined effect of amination and steam explosion pretreatment has the ability to rapidly degrade the lignocellulosic components of rice straw and accelerate the subsequent SSF process.
In fact, they showed that the true protein content by combined ammonification and steam explosion pretreatment of rice straw during 96 h solid-state fermentation was 46.7% of its total matter, rendering it a suitable alternative to high-protein animal feed.
The experiment was balanced with a sufficient number of rice straw samples. The methods of analysis are good, clear and well detailed. The manuscript was well written.
Line 115: I suggest:
‘‘with the aim of’’ instead of ‘‘with the goal’’
Line 133: Put the time of even stirring and impregnation.
Line 176: acide indole acétique (AIA) in French, in English it is indole acetic acid (IAA).
Line 273: ‘‘(p ≤ 0.05)’’ instead of ‘‘(p < 0.05)’’
Line 284: You say that the soluble reducing sugar and nitrogen contents also increased substantially compared to the TC sample (9.4 and 2.5 times, respectively). How did you find these values (9.4 and 2.5)?
Line 288: Have you done any statistical analysis on the sample analysis data after the pre-treatments? If yes, indicate it in the tables by adding the letters (a, b, c ...) on values and the p-value in a new colon.
Line 297: What does the number ‘‘15’’ indicate here?
Line 410-412: Specify fermentation time.
Table 1, 2 and 5: If you have done statistical analysis, indicate it in the tables by adding the letters (a, b, c ...) on values and the p-value in a new colon. Indicate the meaning of the letters (a, b, c......) below the tables.
Table 3: Add a column for p-values. Indicate the meaning of the letters (a, b, c......) below the tables.
Table 4: Put the letters a, b, c and d of the statistical analyses uniformly on the standard errors. Indicate the meaning of the letters (a, b, c......) below the tables.

Round 2
Reviewer 1 Report
The article was well-prepared by the comments. I recommend this document to publish.